# Identification of Chemical Markers for the Discrimination of Radix *Angelica sinensis* Grown in Geoherb and Non-Geoherb Regions Using UHPLC-QTOF-MS/MS Based Metabolomics

**DOI:** 10.3390/molecules24193536

**Published:** 2019-09-30

**Authors:** Kaixue Zhang, Menglin Yan, Shu Han, Longfei Cong, Liyao Wang, Liu Zhang, Lili Sun, Haiying Bai, Guanhua Wei, Hong Du, Min Jiang, Gang Bai, Zhigang Yang

**Affiliations:** 1School of Pharmacy, Lanzhou University, Lanzhou 730000, China; zhangkx2018@lzu.edu.cn (K.Z.); lywang17@lzu.edu.cn (L.W.); zhangliu17@lzu.edu.cn (L.Z.); sunll2018@lzu.edu.cn (L.S.); baihy14@lzu.edu.cn (H.B.); weigh2018@lzu.edu.cn (G.W.); 2College of Pharmacy, Nankai University, State Key Laboratory of Medicinal Chemical Biology, Tianjin 300350, China; 15736875593@163.com (M.Y.); 2120181290@mail.nankai.edu.cn (L.C.); minjiang@nankai.edu.cn (M.J.); gangbai@nankai.edu.cn (G.B.); 3School of Chinese Materia Medica, Beijing University of Chinese Medicine, Beijing 100029, China; hs361015@163.com (S.H.); duhong@vip.163.com (H.D.)

**Keywords:** *Angelica sinensis*, geoherb region, chemical markers, content, UHPLC-QTOF-MS/MS, metabolomics, anti-inflammation, calcium antagonists

## Abstract

This research aimed to discover chemical markers for discriminating radix *Angelica sinensis* (RAS) from different regions and to explore the differences of RAS in the content of four active compounds and anti-inflammatory activities on lipopolysacchride (LPS)-induced RAW264.7 cells and calcium antagonists on the HEK 293T cells of RAS. Nine compounds were selected as characteristic chemical markers by ultra-high-performance liquid chromatography-quadrupole/time-of-flight mass spectrometry (UHPLC-QTOF-MS/MS), based on metabolomics, in order to rapidly discriminate RAS from geoherb and non-geoherb regions. The contents of senkyunolide I and butylidenephthalide in geoherb samples were higher than those in non-geoherb samples, but the contents of ferulic acid and levistolide A were lower in the geoherb samples. Furthermore, the geoherbs showed better nitric oxide (NO) inhibitory and calcium antagonistic activities than the non-geoherbs. These results demonstrate the diversity in quality of RAS between geoherbs and non-geoherbs.

## 1. Introduction

The radix of *Angelica sinensis* (Oliv.) Diels (RAS) has played an important role in traditional Chinese medicine (TCM) for thousands of years, due to its functions of replenishing and invigorating the blood, stopping pain, and moistening the intestines [1]. Phthalides and their dimers, phenolic acids, polysaccharides, and flavonoids are reported as the main constituents of RAS [2]. It was confirmed that RAS can be used to treat inflammation [3], cancer [4], cardiovascular disease [5], and Alzheimer’s disease [6]. The Min County of Gansu Province is the traditional geoherb region of RAS, due to its superior qualities, and RAS from this region is popularly used in clinical practice [7].

Metabolomics, a new branch of systems biology, is a powerful tool for the comprehensive profiling and comparison of metabolites. Liquid chromatography–mass spectrometry (LC–MS) and gas chromatography–mass spectrometry (GC–MS) are the routine analysis methods in metabolomics studies. It was reported that β-ocimene, α-pinene, 3-methylbutanal, heptanes, and butanal are potential markers to distinguish RAS between geoherb and non-geoherb regions, using automated headspace solvent-free micro-extraction/gas chromatography–mass spectrometry (HS-SFME/GC–MS) [8]. Wang et al. found eighteen metabolites in the growth periods of RAS using ultra-high-performance liquid chromatography/time-of-flight mass spectrometry (UHPLC-TOF-MS) [9]. Ten major bioactive components were reported as chemical markers with the effect of different drying methods of RAS using UHPLC-QTOF-MS/MS [10]. (3Z)-6-sulfite-ligustilide and (3E)-6-sulfite-ligustilide are potential characteristic chemical markers for inspecting sulfur-fumigated RAS among commercial RAS samples by UHPLC-QTOF-MS/MS [11]. Li et al. found nine potential metabolite biomarkers in the plasma and nine potential metabolite biomarkers in spleen homogenates on the hematopoietic function of RAS in a blood-deficiency mouse model by GC–MS [12]. To our knowledge, RAS from geoherbs is sold at high prices and has a similar appearance in TCM markets. Meanwhile, studies on the discrimination of RAS between geoherbs and non-geoherbs by LC–MS are still rare, although there are many reports on RAS that were based on metabolomics. Therefore, it is necessary to find a method to rapidly identify geoherb RAS.

There were many recent studies on the determination of the content in RAS samples. The content of ferulic acid was highest in samples collected in Yunnan, followed by the Gansu and Sichuan Provinces [13]. Shi et al. simultaneously determined the content of seven active components in RAS, before and after sulfur fumigation, by UHPLC, and concluded that sulfur fumigation decreased the content of ferulic acid, ligustilide, n-butylidenephthalide, senkyunolide A, senkyunolide H, senkyunolide I, and levistilide A [14].

Inflammatory processes are normal physiological immune functions [15], and many inflammatory disorders exist, such as infection, metabolic diseases, cancers, and aging, in which excessive inflammation occurs [16]. Macrophages, which are essential immune cells, induce inflammation by producing NO, prostaglandin E2, and pro-inflammatory cytokines. The anti-inflammatory mechanism of RAS is speculated to directly or indirectly inhibit target cells secreting TNF-α, IL-6, IL-lβ, IL-2, and NO, promoting anti-inflammatory cytokine release [17]. Li et al. found that essential oil extracted from RAS had a certain inhibitory effect on early and late inflammatory reactions, as well as showing inhibitory activities on the writhing reaction of mice caused by chemical substances [18].

Hypertension is one of the major chronic diseases which causes serious damage to human health. Calcium channel blockers are commonly used to treat hypertension because of their remarkable efficacy [19]. RAS, with the function of promoting blood circulation and removing stasis, could be used as a treatment for hypertension in the elderly. Furthermore, the extracts of RAS could reduce blood pressure through certain receptors and channels, reducing renin–angiotensin activity and lowering lipids and viscosity in modern pharmacology [20]. The volatile oil of RAS could inhibit the contraction of vascular smooth muscle induced by potassium chloride. Its antihypertensive effects on hypertension-model mice was demonstrated, and its inhibitory effects on both receptor-operated Ca^2+^ channels and voltage-operated Ca^2+^ channels mediating aortic smooth muscle contractions were proved [21].

In this study, the UHPLC-QTOF-MS/MS method, based on metabolomics, was applied to comprehensively characterize the chemical components of RAS, in order to find characteristic chemical markers between geoherbs and non-geoherbs. In order to explore the differences in content and activity further, we quantified four active compounds and evaluated their anti-inflammatory activities on LPS-induced RAW264.7 cells and calcium antagonists on the HEK 293T cells.

## 2. Results and Discussion

### 2.1. Characteristic Chemical Markers Analysis by UHPLC-QTOF-MS/MS

#### 2.1.1. Multivariate Data Analysis

In order to examine the chemical differences between geoherbs and non-geoherbs, an unsupervised pattern recognition of the principal component analysis (PCA) was operated. The PCA scores (Figure 1a) show an obvious separation. The samples from the non-geoherb regions presented relative dispersion, which could be related to the fact that they came from distinct geographical regions.

In partial least squares–discriminant analysis (PLS–DA) modelling (Figure 1b), the samples from geoherb and non-geoherb regions were sorted into two groups. It is obvious that the composition of RAS from geoherb regions was distinctively different from that of those from non-geoherb regions. The result was basically consistent with that of the PCA. Environmental factors, including altitude, sunlight, soil, water, temperature, and topography, all have a great influence on the qualities of geoherbs [21]. Consequently, the relationship between quality of RAS and geographical factors can be studied in the future. The R^2^X, R^2^Y, and Q^2^Y of the PLS–DA model were 0.444, 0.972, and 0.442, respectively. The permutation result validated the stability and reliability of this PLS–DA model.

It was possible to select markers that contributed significantly to the grouping by setting the Variable Importance for Projection (VIP) value greater than 1.5 and the *p*-value less than 0.05 in the Moderated *t*-Test. A total of nine markers, **M1**–**M9**, were, therefore, obtained (see Figure 1c and Table 1), and they were highlighted in the PCA loadings (Appendix A). 

It can be seen from the heatmap that most characteristic chemical markers had low content in the non-geoherbs. It was interesting that the content of the marker ions *m*/*z* 130.0868, 313.1074, 387.1074, 563.1888, 641.2017, 751.2359, and 1067.2065 were lower in non-geoherbs and that the content of marker ions *m*/*z* 131.0490 and 163.0752 were lower in geoherbs. Furthermore, the content of **M1** (*m*/*z* 387.1074, Rt 5.523 min) was particularly high in samples from Min County, while its content was significantly lower in the non-geoherbs. Therefore, *m*/*z* 387.1074 was selected as the most important compound for discriminating RAS between geoherbs and non-geoherbs.

To confirm the existence of **M1** in the RAS samples and compare the intensity of the peak in RAS from geoherbs and non-geoherbs, the typical ion at *m*/*z* 387.1074 in positive mode was selected as the diagnostic ion for extraction ion analysis. In the geoherb region samples, the intensity of the peaks at *m*/*z* 387.1074 in positive mode reached up to 10^5^ (Appendix A).

**M7** showed a protonated molecular ion [M + H]^+^ at *m*/*z* 313.1074 and a molecular ion [M + H − H_2_O]^+^ at *m*/*z* 177.0547. Its MS/MS spectrum gave fragment ions at *m*/*z* 145.0285 and 117.0341, which are the characteristic fragment ions of ferulic acid. Therefore, **M7** was tentatively assigned as ferulate [22]. **M9** was tentatively assigned as glutamine, based on the fragment ions at *m*/*z* 130.0868 and 84.0808 [23].

#### 2.1.2. Identification of Major Compounds Detected in RAS

As shown in Figure 2 and Table 2, there were 30 peaks in the mixed solution of all RAS samples, 16 of which were unambiguously or tentatively identified. The structures of eight compounds—chlorogenic acid (3), ferulic acid (4), senkyunolide I (6), senkyunolide A (14), butylphthalide (16), butylidenephthalide (17), z-ligustilide (18), and levistolide A (30)—were identified by comparing their accurate masses and retention times with those of the standard compounds. Peaks 1, 2, 5, 7, 8, and 26–28 were also tentatively assigned by comparison of their MS/MS data with the database or the literature [22]. The structures of the main compounds in RAS are listed in Figure 3, and the MS/MS spectra of compounds detected in RAS are presented in Appendix A.

The MS of compound **1** showed a protonated molecular ion [M + H]^+^ at *m*/*z* 205.0972 and [M + Na]^+^ at *m*/*z* 227.0787 in positive mode. Its MS/MS gave an abundant product ion [M + H − NH_3_]^+^ at *m*/*z* 188.0707 and a weak product ion at *m*/*z* 170.0601 [M + H − NH_3_ − H_2_O]^+^. Its MS/MS also showed a fragment ion at *m*/*z* 118.0654, which indicated that it has an indole group. Furthermore, its t_R_ (3.497 min) suggested that it has relatively high hydrophilicity. Therefore, compound **1** was tentatively assigned as tryptophan [22].

Compound **2** produced an abundant protonated molecular ion [M + H]^+^ at *m*/*z* 163.0392. Product ions at *m*/*z* 135.0433 [M + H − CO]^+^, 107.0491 [M + H − 2 CO]^+^, and 89.0386 [M + H − 3 CO]^+^ were also observed. The loss of a series of CO peaks is characteristic of coumarin compounds, and 117.0337 [M + H − CO − H_2_O]^+^ suggested the presence of one hydroxyl. Thus, compound **2** was tentatively assigned as umbelliferone, which was consistent with the data of the MassBank database (https://massbank.eu/).

Compound **3** presented a protonated molecular ion [M + H]^+^ at *m*/*z* 355.1028 in the positive mode. Thus, its molecular mass was inferred to be 354 Da, which implicated an empirical molecular formula of C_16_H_18_O_9_. Its MS/MS gave fragment ions at *m*/*z* 163.0387 [M + H − C_7_H_12_O_6_]^+^ and 135.0442 [M + H − C_7_H_12_O_6_ − CO]^+^. It was seen to be identical to chlorogenic acid, by comparison with the standard compound.

Compound **4** showed a molecular ion [M + H − H_2_O]^+^ at *m*/*z* 177.0550. It also yielded fragment ions at *m*/*z* 149.0612 [M + H − H_2_O − CO]^+^. Its MS/MS gave a fragment ion at *m*/*z* 117.0334 [M + H − H_2_O − C_2_H_4_O_2_]^+^, and a product ion at *m*/*z* 89.0385 was obtained by the loss of CO of 117.0334. It was determined to be identical to ferulic acid by the characteristic fragment ions and the same retention time as the standard compound.

Compound **5** showed a molecular ion [M + H − H_2_O]^+^ at *m*/*z* 499.1269, and a characteristic fragment ion 163.0386 was C_9_H_7_O_3_. Thus, compound **5** was tentatively assigned as dicaffeoylquinic acid, by referring to data in the literature [22].

Compounds **6** and **7** both showed [M + H − H_2_O]^+^ ion at *m*/*z* 207. The MS/MS spectrum gave fragment ions at *m*/*z* 189 [M + H − 2 H_2_O]^+^ and 165 [M + H − H_2_O − CO − CH_2_]^+^. They were identified as senkyunolide I and senkyunolide H, respectively, by comparing their published MS/MS data [22].

Compound **14** was assigned as senkyunolide A, based on a protonated molecular ion [M + H]^+^ at *m*/*z* 193.1221 and fragment ions at *m*/*z* 175.1105 [M + H − H_2_O]^+^, 147.1162 [M + H − H_2_O − CO]^+^, and 119.0848 [M + H − H_2_O − 2 CO]^+^.

Compounds **16** and **18** showed a protonated molecular ion [M + H]^+^ at *m*/*z* 191. Their MS/MS gave product ions at *m*/*z* 173 [M + H − H_2_O]^+^, 145 [M + H − H_2_O − CO]^+^, and 117 [M + H − H_2_O − 2 CO]^+^. Therefore, compounds **16** and **18** were identified as butylphthalide and Z-ligustilide, respectively, by comparison with the standard compounds.

Compound **17** was identified as butylidenephthalide, based on a protonated molecular ion [M + H]^+^ at *m*/*z* 189.0912 and fragment ions at *m*/*z* 171.0800 [M + H − H_2_O]^+^, 153.0695 [M + H − 2 H_2_O]^+^, and 143.0868 [M + H − H_2_O − CO]^+^.

Compounds **26**–**28** all showed a protonated molecular ion [M + H]^+^ at *m*/*z* 381 in the positive ion mode, and their MS/MS all gave the same fragment ions at *m*/*z* 191 and 173 as ligustilide. Thus, they were tentatively assigned as ligustilide dimers [22].

Compound **30** presented a protonated molecular ion [M + H]^+^ at *m*/*z* 381.2064, and its MS/MS gave fragment ions at *m*/*z* 191.1063 [M + H − ligustilide]^+^, 173.0962 [M + H − ligustilide − H_2_O]^+^, and 145.1009 [M + H − ligustilide − H_2_O − CO]^+^. Thus, compound **30** was identified as levistolide A, by comparison with the standard compound.

### 2.2. Determination of Four Active Compounds in RAS

It was reported that ferulic acid has antidiabetic, hepatoprotective, anticancer, anti-apoptotic, and anti-aging properties [24]. Phthalides are considered to be major bioactive compounds, possessing various activities with anti-tumor, neuroprotective, nephroprotective, analgesic, and anti-angiogenic effects [25]. Therefore, ferulic acid, senkyunolide I, butylidenephthalide, and levistolide A were determined as chemical compounds, which may be used for the quality assessment of RAS, due to their favorable bioactivities.

As shown in Table 3, the average content of ferulic acid in non-geoherbs (samples 1–27) was higher than that in geoherbs (samples 28–40). The result was basically consistent with that of [13]. The average content of levistolide A in samples from non-geoherbs was also much higher than that in geoherbs. Here, we have demonstrated that both senkyunolide I and butylidenephthalide were higher in geoherbs than in non-geoherbs.

### 2.3. Anti-Inflammatory Activity

As shown in Figure 4, the anti-inflammatory activities of RAS from different regions were evaluated by NO production. The average NO inhibition rate of samples from the geoherb regions was 41.53%, and the average of NO inhibition rate of samples from the non-geoherb regions was 39.93%, at a concentration of 5 μg/mL. The *p*-value was less than 0.05 between the two groups, as determined by an independent sample *t*-test (IBM SPSS Statistics 22), which indicated that the average NO inhibition rate of samples from geoherb regions was higher than that of samples from non-geoherb regions. Moreover, the NO inhibition rates of samples from the geoherb regions were better than the positive control (quercetin). Sample 25, from Zhang County, showed the highest NO inhibitory activity. In the future, the sample number can be expanded to study RAS anti-inflammatory activities.

### 2.4. Calcium Antagonistic Activity

The calcium antagonistic activities of RAS extracts were assessed using Ca^2+^ inhibition rates, and the results are shown in Figure 5. The average Ca^2+^ inhibition rates in the geoherb region samples and the non-geoherb region samples were 39.28% and 38.22%, respectively, at a concentration of 10 μg/mL. Samples from geoherbs exerted a higher Ca^2+^ inhibition activity than samples from non-geoherbs, and a significant difference was found between them (*p* < 0.05). Sample 32, from Min County, had the highest Ca^2+^ inhibition rate, and sample 5, from Huzhu County, had the lowest Ca^2+^ inhibition rate. The Ca^2+^ inhibition rates of the samples from geoherbs varied greatly. Similarly, the sample number could be increased, in the future, to further study the relationship between Ca^2+^ inhibition activity and region.

## 3. Experimental

### 3.1. Chemicals and Reagents

MS-grade acetonitrile and MS-grade methanol were purchased from Mreda (Beijing, China). MS-grade water and MS-grade formic acid were obtained from Merck (Darmstadt, Germany). Eight standard compounds, ferulic acid, z-ligustilide, butylphthalide, butylidenephthalide, Chlorogenic acid, levistilide A, senkyunolide I, and senkyunolide A, were purchased from Chengdu Pufei De Biotech Co., Ltd (Chengdu, China).

### 3.2. Plant Materials

A total of 40 samples of RAS were collected from the Gansu, Qinghai, Hubei, and Yunnan Provinces of China. Detailed information is listed in Appendix A. All of the plant materials were identified as Radix *Angelica sinensis* by the author Zhigang Yang. 

### 3.3. UHPLC-QTOF-MS/MS Analysis

#### 3.3.1. Sample Solutions

All samples (100 mesh) were weighed (0.2 g) and extracted with 20.0 mL of 70% methanol in an ultrasonic cleaner at room temperature for 30 minutes. After standing for one hour, the supernatant was centrifuged and filtered at 5000 rpm for 2 min. The filtrates were stored in a refrigerator at 4 °C, for later analysis.

#### 3.3.2. Standard Compound Solutions

The standard compounds were accurately weighed and dissolved by methanol to the final concentration of 10 μg/mL.

#### 3.3.3. Liquid Chromatography Conditions

UHPLC analysis was performed using an Agilent 1290 InfinityⅡequipped with a quaternary pump system and an auto-sampler. The chromatography separation was achieved with a Waters CORTECS UPLC C_18_ column (2.1 mm × 100 mm, 1.6 μm) and maintained at 35 °C throughout the run. The mobile phases were (A) 0.1% formic acid in water and (B) 0.1% formic acid in acetonitrile. The UHPLC elution conditions were optimized, as follows: 0 min 97% A, 2 min 95% A, 5.5 min 50% A, 12 min 35% A, 13 min 0% A, 16 min 0% A, 16.1 min 97% A, and 18 min 97% A, with a flow rate of 0.4 mL/min. The injection volume was 2 μL.

#### 3.3.4. Mass Spectrometry Conditions

The Agilent 6560 Q-TOF mass spectrometer was equipped with an Agilent jet stream electrospray source. Mass spectra were acquired in positive ionization mode with a scan range of 50–1700 Da. The mass spectrometer was operated with the following parameters: nebulizer at 20 psig, capillary voltage at 3500 V, drying gas at 225 °C, drying gas flow at 5 L/min, sheath gas at 400 °C, sheath gas flow at 12 L/min, nozzle voltage at 500 V, and data acquisition at the rate of 1 spectrum/s. MS/MS spectra were used to obtain fragments and identify compounds. The mass spectrometry was tuned to meet the detection requirements of compounds at an accuracy of ±2 ppm before the analysis. Two reference masses (121.050873 and 922.009798) were selected for the autocalibration throughout the run.

#### 3.3.5. Data Processing

All raw data, collected by the Agilent Data Acquisition software (version B.08.00), were imported into the Profinder software (version B.08.00) in the early stage. The samples were grouped by geoherb or non-geoherb region. Choosing the wizard of the batch recursive feature extraction (small molecules/peptides) and following the wizard step by step, the main parameters were listed, as follows. Peaks filters: use peaks with height ≥500 counts; binning and alignment: RT tolerance = 0.00% ± 0.1 min; mass tolerance = ± 10.00 ppm + 2.00 mDa; molecular feature extraction (MFE) and extraction of ion chromatograph (EIC) filters: score ≥80% and 100%, respectively, of the file in at least one sample group in minimum filter matches. The absolute heights in the MFE and EIC filters were 500 and 8000, respectively. The processed data were exported to CEF format and entered into the Agilent Mass Profiler Professional software (MPP, version 14.9, Agilent, Palo Alto, CA, USA), for later analysis.

In MPP, all samples were grouped by geoherb or non-geoherb region. The filter parameter of retaining entities that appeared in at least 100% of the samples, in at least one condition, was set. Then, unsupervised pattern recognition the PCA and supervised pattern recognition the PLS–DA were performed to analyze the intrinsic variations of the data. The characteristic chemical markers were selected by VIP value > 1.5 and *p*-value < 0.05 in the Moderated *t*-Test.

### 3.4. Determination of Four Active Compounds in RAS

#### 3.4.1. RAS Sample Solutions

An accurately weighed 0.5 g of RAS powder was introduced into a 50 mL volumetric flask and 50 mL 70% methanol was added. The volume was made up to 70% methanol after ultrasonic treatment at room temperature for 30 min. The extract was filtered with a 0.22 μm membrane filter. An aliquot of 4 μL of solution was used as the injection in the UHPLC analysis.

#### 3.4.2. Preparation of Standard Solutions for Linearity and Calibration

An accurately weighed 1.46 mg of ferulic acid standard compounds was introduced into a 25 mL volumetric flask and was made up to volume with methanol; this was used as calibration solution one. Accurately weighed 2.91, 7.37, and 2.0 mg of levistilide A, butylidenephthalide, and senkyunolide I standard compounds, respectively, were separately transferred into a 50 mL volumetric flask, and methanol was added up to volume; this was used as calibration solution two. The samples were monitored at a wavelength of 276 nm for levistilide A and senkyunolide I, and 322 nm for ferulic acid and butylidenephthalide. Peak areas against content were plotted to obtain the calibration curves of the standard compounds. The R^2^ value of each standard compound was higher than 0.999 in the linear range.

#### 3.4.3. Liquid Chromatography Conditions

The contents of seven active compounds were determined using an Agilent 1290 InfinityⅡ consisting of a quaternary pump system and an auto-sampler. Chromatography separation was achieved with a Waters CORTECS UPLC C_18_ column (2.1 mm × 100 mm, 1.6 μm) and maintained at 35 °C during the run. The mobile phases were (A) 1% formic acid in water and (B) acetonitrile. A linear solvent gradient of A-B was optimized as follows: 0 min 93% A, 0.8 min 90% A, 3.5 min 75% A, 5.5 min 56% A, 9.1 min 47% A, 9.6 min 35% A, 10.6 min 32% A, 11.6 min 32% A, and 11.61 min 0% A, with a flow rate of 0.4 mL/min. The injection volume was 4 μL. The detection wavelength of DAD (Diode Array Detection) was 190–400 nm.

### 3.5. Anti-Inflammatory Activity

#### 3.5.1. Extracts of RAS

RAS powders (100 mesh) were weighed (to approximately 5.0 g) and extracted with 40.0 mL of 80% ethanol, followed by sonication two times. The obtained extracts were mixed together and allowed to concentrate in a rotary evaporator. The residue was dissolved in methanol.

#### 3.5.2. Cell Culture

The RAW264.7 macrophage cell line was purchased from National Infrastructure of Cell Line Resource. Cells were cultured at 37 °C/5% CO_2_ in Dulbecco’s Modified Eagle Medium (DMEM, 10% fetal bovine serum, 1% penicillin, and streptomycin). DMEM was replaced every two days, and cells were allowed to subculture when they reached 80%–90% confluency.

#### 3.5.3. NO Determination

RAW264.7 cells were plated in 96-well culture plates and followed four different treatments—DMSO (negative group), LPS (1 μg/mL, model group), LPS + quercetin (5 μg/mL, positive group), and LPS + extracts of RAS (5 μg/mL, treatment group)—for NO determination. Nitrite (NO_2_^−^) in the culture medium was measured as an indicator of NO production, using the Griess reaction. All agents were added at the same time, and the groups were treated for 24 h. After treatment with the same concentrations of extracts of RAS and quercetin, the supernatant of the cells was mixed with an equal volume of Griess reagent, and absorbance of the mixture was measured at 540 nm. The experiments were repeated three times independently.

### 3.6. Calcium Antagonistic Activity

#### 3.6.1. Cell Culture

The HEK 293T cells were purchased from the American Type Culture Collection (Rockville, MD, USA) and were cultured at 37 °C/5% CO_2_ in DMEM (high glucose) with 10% fetal bovine serum and 1% streptomycin–ampicillin.

#### 3.6.2. Cell Administration

The HEK 293T cells were plated in 96-well plates and co-transfected with the Ca^2+^ luciferase reporter plasmid PGL 4.30 (100 ng/well) and a Renilla plasmid (10 ng/well) in the serum-free medium when the degree of cell confluence reached 50%–70%. Then, serum-containing DMEM (high glucose) was added. Transfection was performed for 22 h using PEI liposome 2000 (1 mg/mL) as a transfection reagent. The experiment was operated using different treatments of blank medium (negative group), 1 × 10^−3^ mol/L ionomycin and 1 mg/mL phorbol ester (model group), verapamil (1 × 10^−5^ mol/L, positive group), and extracts of RAS (1 × 10^−5^ kg/L, treatment group). After six hours of administration, the cells were collected for gene reporter detection, according to the ratio of the relative fluorescence intensity (Ca^2+^ fluorescence value/Renilla fluorescence value). The experiments were repeated three times independently.

## 4. Conclusions

A total of 37 compounds were detected, and nine characteristic chemical markers were selected for discriminating RAS from different regions by UHPLC-QTOF-MS/MS. The average contents of senkyunolide I and butylidenephthalide in geoherb samples were higher, but the average contents of ferulic acid and levistolide A were lower than those in non-geoherb samples. Meanwhile, the geoherbs showed higher anti-inflammatory and calcium-antagonistic activities than the non-geoherbs. Significantly, **M1**, with the typical ion at *m*/*z* 387.1074 in positive mode, made the greatest contribution to the grouping and, so, could be used as a diagnostic ion in extraction ion analysis. These results indicated that RAS from geoherb regions and non-geoherb regions showed diversity, and it is possible to discriminate RAS from different regions rapidly. 

## Figures and Tables

**Figure 1 molecules-24-03536-f001:**
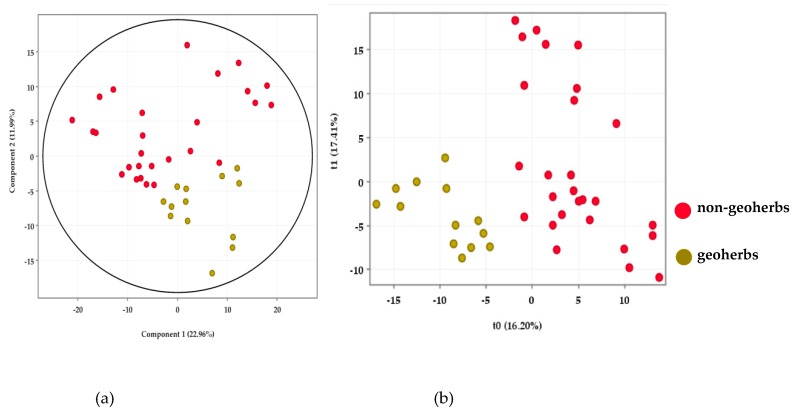
PCA scores (**a**), PLS–DA scores (**b**), and heatmap of nine characteristic chemical markers (**c**).

**Figure 2 molecules-24-03536-f002:**
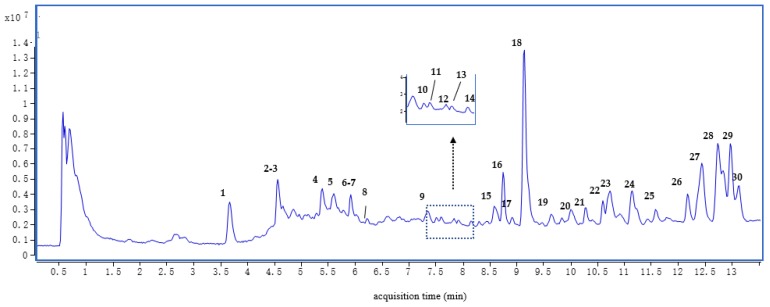
Total ion of chromatography (TIC) in a mixed solution of all RAS samples.

**Figure 3 molecules-24-03536-f003:**
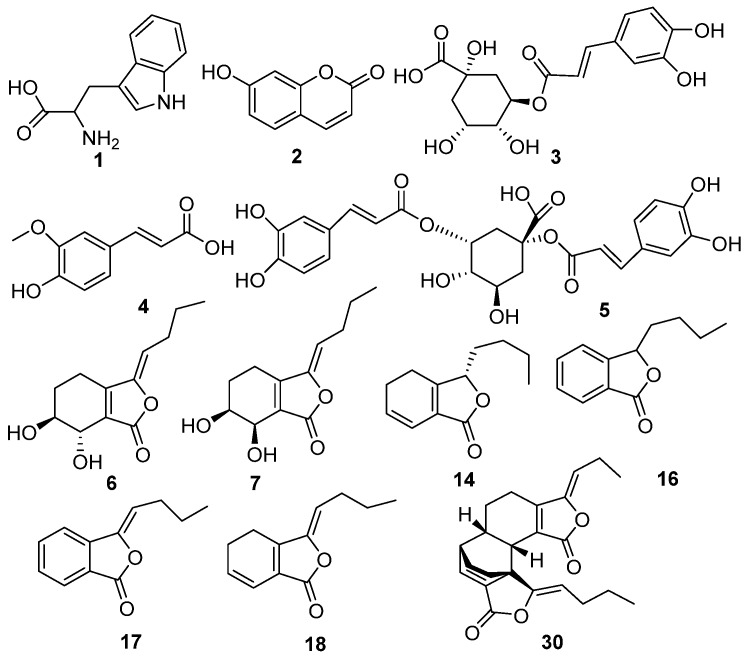
Structures of main compounds in RAS.

**Figure 4 molecules-24-03536-f004:**
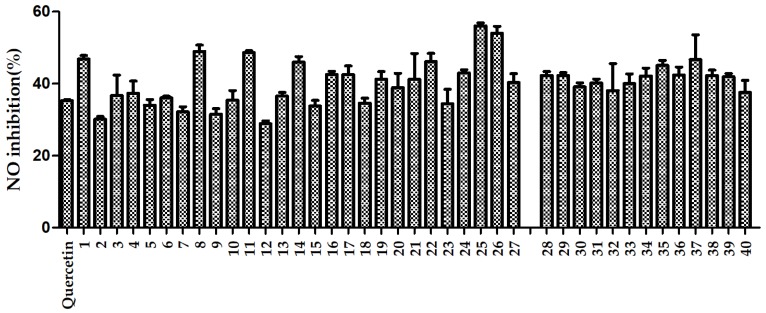
NO inhibition rates of extracts of RAS from non-geoherbs (1–27) and geoherbs (28–40).

**Figure 5 molecules-24-03536-f005:**
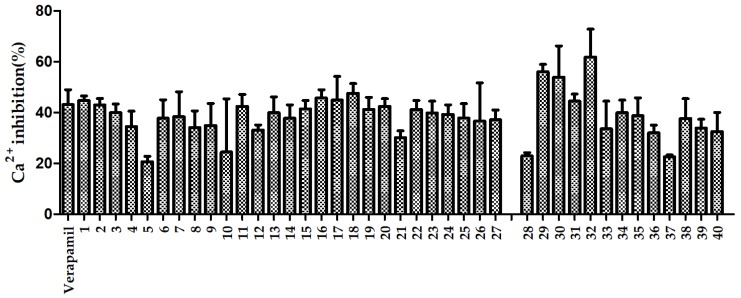
Ca^2+^ inhibition rates of extracts of RAS from non-geoherbs (1–27) and geoherbs (28–40).

**Table 1 molecules-24-03536-t001:** Tentative markers for discriminating radix *Angelica sinensis* (RAS) from geoherbs and non-geoherbs.

Compound	*m*/*z* (ESI^+^)	Rt (min)	Tentative Formula	VIP	MS/MS Fragment Ion (*m*/*z*)	Identification
**M1**	387.1074	5.523	C_20_H_18_O_8_	8.739	371.2273, 283.1760, 177.1127, 133.0857, 89.0597, 45.0338	Unknown
**M2**	563.1888	6.583	C_38_H_26_O_5_	2.184	481.2603, 305.1541, 207.1018, 133.0865, 89.0597, 45.0337	Unknown
**M3**	163.0752	7.600	C_10_H_10_O_2_	2.032	131.0476, 103.0546, 77.0385	Unknown
**M4**	641.2017	6.976	C_36_H_32_O_11_	1.923	323.0893, 291.0971, 83.0855, 45.0337	Unknown
**M5**	131.0490	7.598	C_6_H_10_O_3_	1.908	115.0535, 103.0538	Unknown
**M6**	751.2359	8.176	C_24_H_46_O_26_	1.860	557.1798, 395.1120, 163.0749, 131.0493, 103.0535	Unknown
**M7**	313.1074	6.218	C_18_H_16_O_5_	1.678	177.0547, 145.0285, 117.0341	Ferulate
**M8**	1067.2065	5.609		1.565	551.0790	Unknown
**M9**	130.0868 [M + H − NH_3_]^+^	0.802	C_5_H_10_N_2_O_3_	1.536	84.0808, 56.0497	Glutamine

**Table 2 molecules-24-03536-t002:** Mass data of compounds detected in RAS.

Peak	Identification	Rt (min)	Formula	*m*/*z* (ESI^+^)	MS/MS Fragment Ion (*m*/*z*)
**1**	Tryptophan	3.497	C_11_H_12_N_2_O_2_	188.0707 [M + H − NH_3_]^+^	170.0601, 118.0654
**2**	Umbelliferone	4.547	C_9_H_6_O_3_	163.0392	135.0433, 117.0337, 107.0491, 89.0386
**3**	Chlorogenic acid	4.568	C_16_H_18_O_9_	355.1028	163.0387, 135.0442
**4**	Ferulic acid	5.487	C_10_H_10_O_4_	177.0550 [M + H − H_2_O]^+^	149.0612, 117.0334, 89.0385
**5**	Dicaffeoylquinic acid	5.549	C_25_H_24_O_12_	499.1269 [M + H − H_2_O]^+^	163.0386
**6**	Senkyunolide I	5.893	C_12_H_16_O_4_	207.1015 [M +H − H_2_O]^+^	189.0909, 165.0549
**7**	Senkyunolide H	5.948	C_12_H_16_O_4_	207.0991 [M + H − H_2_O]^+^	189.0893, 165.0538
**8**	Ferulate	6.218	C_18_H_16_O_5_	313.1074	177.0547, 145.0285, 117.0341
**9**	Unknown	7.375	C_18_H_36_O_3_	318.2994 [M + NH_4_]^+^	256.2620, 88.0757
**10**	Unknown	7.532	C_16_H_22_O_4_	279.1590	191.1075, 105.0325, 71.0493
**11**	Unknown	7.628	C_10_H_10_O_2_	163.0756	131.0476, 103.0546, 77.0385
**12**	Unknown	7.850	C_24_H_22_O_7_	423.1418	229.0840, 189.0901
**13**	Unknown	8.006	C_18_H_20_O_5_	316.2849	299.2822, 256.2624, 60.0446
**14**	Senkyunolide A	8.163	C_12_H_16_O_2_	193.1221	175.1105, 147.1162, 119.0848
**15**	Unknown	8.749	C_27_H_46_O_9_	514.3166	355.2881, 184.0728, 100.1122
**16**	Butylphthalide	8.758	C_12_H_14_O_2_	191.1067	173.0959, 145.1008, 117.0697
**17**	Butylidenephthalide	8.942	C_12_H_12_O_2_	189.0912	171.0800, 153.0695, 143.0868
**18**	Z-ligustilide	9.152	C_12_H_14_O_2_	191.1066	173.0964, 145.1015, 117.0701
**19**	Unknown	9.738	C_30_H_47_O_7_	520.3391	337.2703, 184.0733
**20**	Unknown	10.002	C_30_H_47_O_7_	520.3393	337.2733, 184.0732
**21**	Unknown	10.317	C_32_H_50_O_12_	627.3358	541.2505, 465.2823
**22**	Unknown	10.643	C_32_H_50_O_12_	627.3351	465.2816, 447.2724
**23**	Unknown	10.783	C_27_H_43_O_7_	480.3112	100.1123
**24**	Unknown	11.152	C_27_H_43_O_7_	480.3112	100.1123
**25**	Unknown	11.640	C_30_H_50_O_12_	603.3350	441.2839, 423.2710
**26**	Ligustilide dimer	12.208	C_24_H_28_O_4_	381.2064	191.1064, 173.0952
**27**	Ligustilide dimer	12.429	C_24_H_28_O_4_	381.2064	191.1064, 173.0957
**28**	Ligustilide dimer	12.749	C_24_H_28_O_4_	381.2064	191.1068, 173.0963
**29**	Unknown	13.047	C_27_H_43_O_7_	480.3112	100.1120
**30**	Levistolide A	13.152	C_24_H_28_O_4_	381.2064	191.1063, 173.0962, 145.1009

**Table 3 molecules-24-03536-t003:** The contents of four compounds in RAS (mg/g).

Sample No.	Ferulic Acid	Senkyunolide I	Butylidenephthalide	Levistolide A
1	1.0976	0.1481	0.0824	0.0724
2	0.9790	0.1423	0.0892	0.0830
3	0.9266	0.1677	0.1238	0.0746
4	1.6556	0.1075	0.0650	0.0828
5	1.0153	0.2211	0.2026	0.0697
6	1.2751	0.2059	0.1763	0.0856
7	0.8165	0.1905	0.1936	0.0627
8	1.3597	0.2590	0.1804	0.0877
9	1.3231	0.2311	0.1469	0.0985
10	1.6048	0.2470	0.1465	0.1274
11	2.3298	0.3667	0.2347	0.2981
12	2.1577	0.2024	0.1752	0.2405
13	1.1667	0.6794	0.2121	0.2356
14	2.2131	0.2126	0.1818	0.2642
15	1.2855	0.2823	0.1316	0.2611
16	0.8658	0.2767	0.1867	0.9618
17	1.4806	0.2365	0.2191	0.0999
18	0.7851	0.7021	0.2231	0.0646
19	1.8546	0.2832	0.1892	0.1267
20	1.2325	0.1988	0.1337	0.0858
21	1.0423	0.3294	0.2152	0.0788
22	1.3093	0.2704	0.1325	0.0949
23	1.5488	0.3120	0.2103	0.1424
24	0.9638	0.2752	0.1413	0.1096
25	1.6003	0.3040	0.1747	0.1242
26	1.0952	0.3470	0.1576	0.0898
27	1.4920	0.2616	0.1734	0.1294
average	1.3510 ± 0.4210	0.2763 ± 0.1348	0.1666 ± 0.0442	0.1575 ± 0.1749
28	1.4253	0.1977	0.1968	0.0725
29	1.0364	0.1999	0.1919	0.0702
30	0.9771	0.3662	0.1636	0.0807
31	1.0169	0.3201	0.1737	0.0791
32	0.6210	0.3031	0.1243	0.0766
33	1.1375	0.3156	0.1616	0.0926
34	1.0421	0.3486	0.2223	0.0614
35	1.3107	0.3217	0.2163	0.0757
36	1.0917	0.1802	0.1966	0.0721
37	1.5745	0.2259	0.1342	0.1149
38	1.6468	0.2743	0.1868	0.1104
39	0.9111	0.5617	0.2341	0.0678
40	1.3209	0.2343	0.1591	0.0977
average	1.1625 ± 0.2841	0.2961 ± 0.1006	0.1816 ± 0.0330	0.0825 ± 0.0165

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
