# Peer review of "Identification of Chemical Markers for the Discrimination of Radix Angelica sinensis Grown in Geoherb and Non-Geoherb Regions Using UHPLC-QTOF-MS/MS Based Metabolomics"

_molecules, 2019, doi:10.3390/molecules24193536_

Round 1

Reviewer 1 Report

The authors must show IC50 value of any compounds and/or fractions for biological activities

Author Response

Comment: The authors must show IC50 value of any compounds and/or fractions for biological activities.

Response: Thank you very much. We feel sorry that we cannot add the IC50 value in the manuscript, because some financial problems, and we think that the inhibition rate at one concentration could be used to evaluate the biological activities. Recently, some articles published in Molecules also reported biological activities at one concentration of compounds and/or extracts, such as the article titled “Bioactive Constituents from the Aerial Parts of Pluchea indica Less” (Molecules 2018, 23, 2104; https://doi:10.3390/molecules23092104), and the article titled “Tyrosinase Inhibition Antioxidant Effect and Cytotoxicity Studies of the Extracts of Cudrania tricuspidata Fruit Standardized in Chlorogenic Acid” (2019, 24(18), 3266; https://doi.org/10.3390/molecules24183266).

Reviewer 2 Report

The manuscript presents nine chemical markers for discriminating the radix of angelica sinensis in geoherb and non-geoherb regions. It also provides the quantification of four active compounds and the evaluation of the anti-inflammatory and Ca antagonist activity of some RAS components. 

Although the manuscript is interesting, the main aim of this work is not clear. For example, in the abstract it is not clear whether the goal is to identify markers or to assay the activity of RAS active components or to quantify them? The manuscript should be better structured around the main goal of this work.  

The title should be rephrased e.g., the authors can change "based on metabolomics approach" in " by LC-MS based metabolomics".

the introduction reports information not strictly relevant to the aim of the work and should be re-structured to provide context for the main aim. Moreover, in the last two sentences of the introduction the declared aim is different from what reported in the title and the abstract.  Results, multivariate analysis: it is not clear on which data sets the PCA and the PLS-DA was performed. The authors should provide a brief description of the data set/groups of samples used.  Results, "identification compunds of RAS": the title of this paragraph should be rephrased. Compound 1: [M+Na]+ is not a molecular ion, please correct. Compound 2: "which was consistent with the database: which database are you referring to? please add reference.  Results, determination of active compounds of RAS: the second paragraph starts with "as a result" however this concept appears disconnected from the previous one. The title of the manuscript does not refer to this part, however we find it interesting and we suggest to stress it out more in the abstract and title.  Overall the manuscript requires major revisions before acceptance. 

Author Response

Comment: Although the manuscript is interesting, the main aim of this work is not clear. For example, in the abstract it is not clear whether the goal is to identify markers or to assay the activity of RAS active components or to quantify them?

Response: Thanks for your suggestions and questions. The manuscript aims to discover chemical markers for discriminating radix Angelica sinensis (RAS) from different regions and to explore the difference of RAS in contents and activities between geoherbs and non-geoherbs further.

Comment: The title should be rephrased e.g., the authors can change "based on metabolomics approach" in " by LC-MS based metabolomics".

Response: The title has been rephrased as your suggestion. The title has been changed into Characteristic Chemical Markers for the Discrimination of Radix Angelica sinensis between Geoherb and non-Geoherb Regions by UHPLC-QTOF-MS/MS Based on Metabolomics Approach.

Comment: The introduction reports information not strictly relevant to the aim of the work and should be re-structured to provide context for the main aim. Moreover, in the last two sentences of the introduction the declared aim is different from what reported in the title and the abstract. 

Response: The introduction reports information has been reorganized to clarify my aim of the work. Moreover, the last two sentences of the introduction have been corrected and were consistent with those reported in the title and the abstract.

Comment: Results, multivariate analysis: it is not clear on which data sets the PCA and the PLS-DA was performed. The authors should provide a brief description of the data set/groups of samples used.

Response: The data sets of the PCA and the PLS-DA were shown in 3.3.5 Data processing section. The description in the manuscript was The filter parameter of retaining entities that appeared in at least 100.0% of the samples, in at least one condition, was set.

Comment: Results, "identification compunds of RAS": the title of this paragraph should be rephrased. Compound 1: [M+Na]+ is not a molecular ion, please correct. Compound 2: "which was consistent with the database: which database are you referring to? please add reference. 

Response: The title of identification compounds of RAS was replaced by Identification of major compounds detected in the RAS. “a molecular ion” have been deleted. We have also added the database  (https://massbank.eu/) of compound 2.

Comment: Results, determination of active compounds of RAS: the second paragraph starts with "as a result" however this concept appears disconnected from the previous one.

Response: Thanks a lot. We felt sorry for the simple error. We wanted to express the meaning of  as shown in Table 3.

Comment: The title of the manuscript does not refer to the part of determination of active compounds of RAS, however we find it interesting and we suggest to stress it out more in the abstract and title.

Response: Thanks, we have corrected the abstract as your suggestions.

Reviewer 3 Report

The manuscript entitled “Characteristic Chemical Markers for Discrimination of Radix Angelica sinensis between Geoherb Region and non Geoherb Region Based on Metabolomics Approach” reports the metabolic profiling of A. sinensis herb from different cultivation regions using UHPLC-Q/TOF MS. The authors also performed anti-inflammatory and antihypertensive in vitro assays.

According to the literature, A. sinensis has been widely used in popular medicine due to the anti-inflammatory properties. Previous phytochemical studies have also shown the chemical diversity of this medicinal plant. Systematic investigations are important to understand the etnopharmacological potential of A. sinensis, and support the discrimination of adulterants and/or contaminants.

However, some aspects should be improved in order to be accept:

1) The English should be improved. There are redundant words (distinctively different), inappropriate terms, and sections that could be condensed for better understanding.

2) what is the difference between geoherb and non-geoherb? Is it the geographic location? And how this variation would impact on A. sinensis production? It is not clear whether the geographic variation would affect the pharmacological potential or the quality of the plant material.

3) Based on the previous question, what are the main goals of the manuscript? Is the identification of bioactive compounds or the detection of chemical markers for quality control purposes?

4) What is the difference of section 2.1.1 Multivatiate data analysis, and 2.1.3 Tentative markers assignment? The sections should be the same. The authors should plot the scores and loadings for PC1 × PC2, highlighting the detected metabolites. They should also replace the heatmap (figure 4) for another graph. The heatmap works better for large datasets. 

5) The MS/MS data for the detected metabolites should be add in the supplementary material. The authors should remove the stereochemistry of those metabolites putatively identified. Compounds 6 and 7 were described with low-resolution m/z values.

6) The authors should identify compound m/z 387.1074 due to its importance to the metabolite discrimination by geographic condition.

7) What is the relation between the bioassays and the geographic location in terms of chemical composition? Why didn`t the authors perform the bioactivity assay using the standards? The anti-inflammation results looks very similar across all the samples. And why did the authors perform the calcium antagonistic in vitro bioassay?

Author Response

Comment 1: The English should be improved. There are redundant words (distinctively different), inappropriate terms, and sections that could be condensed for better understanding.

Response: We have corrected the manuscript using a professional English editing service from MDPI for better understanding as your suggestion.

Comment 2: What is the difference between geoherb and non-geoherb? Is it the geographic location? And how this variation would impact on A. sinensis production? It is not clear whether the geographic variation would affect the pharmacological potential or the quality of the plant material.

Response: Medicinal material from geoherb region is produced and assembled in specific geographic regions with designated natural conditions and ecological environment, with particular attention to cultivation technique, harvesting and processing. Generally speaking, the biggest difference between geoherbs and non-geoherbs was geographic location. It was agreed that the radix Angelica sinensis (RAS) from geoherb region with superior qualities. The geoherbs showed better nitric oxide (NO) inhibitory activities and calcium antagonistic activities than non-geoherbs in our results.

It was indeed that geographic variation would affect the pharmacological potential or the quality of the medicinal material in the manuscript by comparing the anti-inflammatory activities on lipopolysaccharide (LPS)-induced RAW264.7 cells and calcium antagonistic activities on the HEK 293T cells of RAS between geoherbs and non-geoherbs.

Comment 3: Based on the previous question, what are the main goals of the manuscript? Is the identification of bioactive compounds or the detection of chemical markers for quality control purposes?

Response: The manuscript mainly aims to discover chemical markers for discriminating RAS from different regions, in addition, we also want to explore the difference of RAS in contents and activities between geoherbs and non-geoherbs.

Comment 4: What is the difference of section 2.1.1 Multivatiate data analysis, and 2.1.3 Tentative markers assignment? The sections should be the same. The authors should plot the scores and loadings for PC1 × PC2, highlighting the detected metabolites. They should also replace the heatmap (figure 4) for another graph. The heatmap works better for large datasets.

Response: There was no difference between section 2.1.1 Multivatiate data analysis, and 2.1.3 Tentative markers assignment. These two parts have been merged as the manuscript showed. We have plotted the scores for PC1×PC2 in the manuscript (2.1.1 Multivariate data analysis Figure 1) and loadings for PC1×PC2, highlighting the detected metabolites in the supplement (Figure S1). Furthermore, we could intuitively obtained content level of the nine characteristic chemical markers in geoherbs and non-geoherbs from the heatmap, thus, we decided to remain the heatmap.

Comment 5: The MS/MS data for the detected metabolites should be add in the supplementary material. The authors should remove the stereochemistry of those metabolites putatively identified. Compounds 6 and 7 were described with low-resolution m/z values.

Response: The MS/MS data for the detected metabolites have been added into the supplementary material (Figure S3 ). We used standard compounds to unambiguously identify some compounds, moreover, we only showed main compounds with stereochemistry of RAS in Figure 3. High-resolution m/z values of compound 6 and 7 were listed in the Table 2.

Comment 6: The authors should identify compound m/z 387.1074 due to its importance to the metabolite discrimination by geographic condition.

Response: Thank you very much. We also want to isolate and identify this compound,  however, as you known, the process of traditional methods for extraction and separation of unknown compounds is time-consuming. We would report this compound and other compounds together in future.

Comment 7: What is the relation between the bioassays and the geographic location in terms of chemical composition? Why didn`t the authors perform the bioactivity assay using the standards? The anti-inflammation results looks very similar across all the samples. And why did the authors perform the calcium antagonistic in vitro bioassay?

Response: As bioactivity results showed, RAS from geoherbs and non-geoherbs presents different effect, which might be related to their minor chemical composition.

Our aim was to compare bioactivity of RAS extracts between geoherbs and non- geoherbs, thus, we did not determine the bioactivity assay of standards.

Although the anti-inflammation results looked very similar across all the samples, as we mentioned in the manuscript, the P-value between geoherbs and non-geoherbs was less than 0.05 by independent sample T-test, which means it has statistical significance.

RAS with the function of promoting blood circulation and removing stasis could be used as a treatment for hypertension in traditional Chinese medicine clinical, and calcium channel blockers are commonly used to treat hypertension in modern medicine, so we performed the calcium antagonists in vitro bioassay.

Round 2

Reviewer 2 Report

The authors improved the manuscript as requested. 

We suggest to simplify the title as following:

"Characteristic Chemical Markers for the Discrimination of Radix Angelica sinensis in Geoherb and non-Geoherb Regions by UHPLC-QTOF-MS/MS Based Metabolomics"